# General destabilizing effects of eutrophication on grassland productivity at multiple spatial scales

Yann Hautier ⬤ et al.#

Eutrophication is a widespread environmental change that usually reduces the stabilizing effect of plant diversity on productivity in local communities. Whether this effect is scale dependent remains to be elucidated. Here, we determine the relationship between plant diversity and temporal stability of productivity for 243 plant communities from 42 grasslands across the globe and quantify the effect of chronic fertilization on these relationships. Unfertilized local communities with more plant species exhibit greater asynchronous dynamics among species in response to natural environmental fluctuations, resulting in greater local stability (alpha stability). Moreover, neighborhood communities that have greater spatial variation in plant species composition within sites (higher beta diversity) have greater spatial asynchrony of productivity among communities, resulting in greater stability at the larger scale (gamma stability). Importantly, fertilization consistently weakens the contribution of plant diversity to both of these stabilizing mechanisms, thus diminishing the positive effect of biodiversity on stability at differing spatial scales. Our findings suggest that preserving grassland functional stability requires conservation of plant diversity within and among ecological communities.

---

#A list of authors and their affiliations appears at the end of the paper.

Humans are altering global nutrient cycles via combustion of fossil fuels and fertilizer application[1]. We have more than doubled preindustrial rates of nitrogen (N) and phosphorus (P) supply to terrestrial ecosystems[2]. Terrestrial N and P inputs are predicted to reach levels that are three to four times preindustrial rates by 2050 (ref. [3]). This pervasive global eutrophication will have dramatic consequences on the structure and functioning of terrestrial and aquatic ecosystems[3]. In grasslands, nutrient enrichment usually increases primary productivity, but reduces plant diversity, and alters the ability of ecosystems to reliably provide functions and services for humanity[4–7].

Concerns that eutrophication compromises both the diversity and stability of ecosystems have led to a growing number of theoretical and empirical studies investigating how these ecosystem responses may be mechanistically linked[4,6,8–11]. These studies have repeatedly shown that the positive effect of plant species richness on the temporal stability of community productivity in ambient (unfertilized) conditions is usually reduced with fertilization[4–6]. However, these studies have primarily focused on plant responses at relatively small scales (i.e., within single local communities). Whether fertilization reduces the positive effect of diversity on temporal stability at larger scales (i.e., among neighboring local communities) remains unclear. Filling this knowledge gap is important because the stable provision of ecosystem services is critical for society[12]. This is especially true, given an increasing concern for large variability of environmental conditions due to multiple anthropogenic influences, including eutrophication and climate change[13].

A recent theoretical framework allows the quantification of the processes that determine the stability of ecosystem functioning at scales beyond the single local community (Fig. 1)[14–16]. Stability at any given scale is defined as the temporal mean of primary productivity divided by its standard deviation[17]. Higher local scale community stability (alpha stability) can result from two main processes. First, a higher average temporal stability of all species in the community (species stability) can stabilize community productivity due to lower variation in individual species abundances from year to year (Fig. 1b). Second, more asynchronous temporal dynamics among species in response to environmental fluctuations (species asynchrony) can stabilize community productivity because declines in the abundance of some species through time are compensated for by increases in other species (Fig. 1c). Higher stability at the larger scale (gamma stability) can result from higher alpha stability and more asynchronous dynamics across local communities (spatial asynchrony; Fig. 1d). Thus, the stabilizing effect of spatial asynchrony on productivity at the larger scale (spatial insurance hypothesis)[14,18] mirrors the stabilizing effect of species asynchrony on productivity at the local scale (species or local insurance hypothesis)[8,16,19,20]. Higher species asynchrony and species stability can result from higher local species diversity through higher species richness[9,21,22], higher species evenness[8], or both (e.g., higher values of diversity indices—such as the Shannon index—that combines the two[23]; Fig. 1e). Higher spatial asynchrony can result from greater local species diversity or higher variation in species composition among communities (beta diversity)[16].

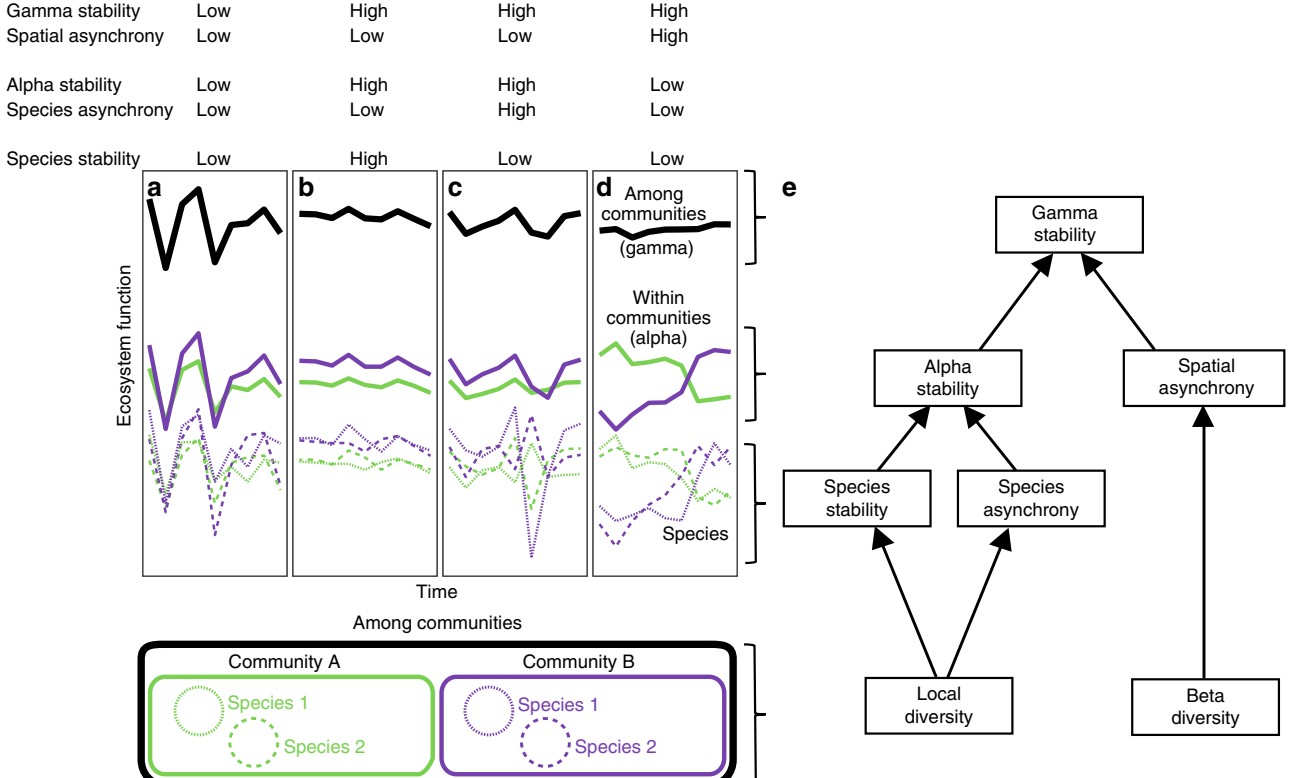

**Fig. 1 Conceptual figure illustrating the nonexclusive processes by which species stability, species asynchrony, and spatial asynchrony may contribute to stabilize functioning (such as productivity) within (alpha stability) and among communities (gamma stability). a** Low stability and asynchrony of species within communities result in low alpha stability that in turn results in low gamma stability under low degree of asynchronous dynamics among communities (spatial asynchrony). Relatively high alpha and gamma stability may result from **b** high species stability and **c** high species asynchrony. **d** Relatively high gamma stability may additionally result from high spatial asynchrony. **e** Path analysis used to assess the relationship of local and beta diversity with the mechanisms promoting stability at multiple spatial scales under unmanipulated control or fertilized condition. Note that species names belong to a given community, they could or could not be the same species among communities. Adapted from Wilcox et al.[21].

**Table 1 Hypotheses related to key predictions from theories relating biodiversity, asynchrony, and stability within and among interconnected communities.**

| Pathway | Hypotheses and mechanisms | References |
|---|---|---|
| **Within communities** | | |
| Species richness → species stability | Higher plant richness within a community either increases or decreases the temporal stability of species abundances within the community by either decreasing or increasing variation in individual species abundances from year to year. | 8 |
| Species richness → species asynchrony | Higher plant richness within a community provides greater likelihood for asynchronous fluctuations among species to compensate one another when the number of species is higher. | 51 |
| Species stability → alpha stability | Higher temporal stability of species abundances within the community increases the temporal stability of community productivity due to lower variation in individual species abundances from year to year | 8,14 |
| Species asynchrony → alpha stability | Higher species asynchronous responses to environmental fluctuations within the community increases the temporal stability of community productivity because declines in the abundance of some species are compensated for by increases in others, thus buffering temporal fluctuation in the abundance of the whole community (species or local insurance hypothesis). | 8,14,19,20 |
| **Among communities** | | |
| Beta diversity → spatial asynchrony | Higher variation and dissimilarity in species composition among communities increase asynchronous community responses to environmental fluctuations. | 16 |
| Alpha stability → gamma stability | Higher temporal stability of local communities cascades to larger scales and increase the temporal stability of total ecosystem function at the landscape level | 15 |
| Spatial asynchrony → gamma stability | Higher asynchronous community responses to environmental fluctuations increase temporal stability of productivity at the larger scale because declines in the productivity of some communities are compensated for by increases in others, thus buffering temporal fluctuation in the productivity of interconnected local communities (spatial insurance hypothesis). | 15,18 |

According to this framework, fertilization can affect the links between diversity, asynchrony, and stability across spatial scales (Fig. 1e and Table 1). At the local scale, fertilization can decrease niche dimensionality, and favor a few dominant plant species by affecting the competitive balance among species, potentially reducing the insurance effects of local diversity[7,22]. At the larger scale, fertilization can reduce spatial heterogeneity in community composition, and decrease variations among local plant community structure, potentially reducing the spatial insurance effect of beta diversity[16]. Moreover, fertilization often reduces plant diversity, which could in turn reduce asynchrony and stability at multiple scales[4,9,17,24]. However, the role of fertilization in mediating the functional consequences of biodiversity changes (variations in the number, abundance, and identities of species) and compensatory mechanisms (variation and compensation in species responses) that can affect the stable provisioning of ecosystem functions at larger spatial scales remains to be elucidated[25].

To our knowledge, only one recent study has assessed the effect of nutrient enrichment on stability within and among interconnected communities in a temperate grassland[26]. By adding different nitrogen treatments to communities in ten blocks spread out within a single site, that study found that 5 years of chronic nitrogen addition reduced alpha stability through a decline in species asynchrony, but had no effect on spatial asynchrony. However, these conclusions were based on a single grassland site manipulating a single nutrient, with the implicit assumption that the relationship between diversity and stability was unaffected by eutrophication. This argues for multisite comparative studies assessing the generality of the mechanistic links between these ecosystem responses to eutrophication.

Here, we use a coordinated, multisite and multiyear nutrient enrichment experiment (±chronic nitrogen, phosphorus, and potassium addition, Nutrient Network (NutNet)[27]) to assess the scale dependence of fertilization impacts on plant diversity and stability. Treatments were randomly assigned to 25 m² plots and were replicated in three blocks at most sites (Supplementary Data 1). Samples were collected in 1 m² subplots across 243 communities from 42 grassland sites on six continents and followed a standardized protocol at all sites[27]. We selected these sites

as they contained between 4 and 9 years of experimental duration (hereafter "period of experimental duration"), and three blocks per site, excluding additional blocks from sites that had more than three (Supplementary Data 1). Sites spanned a broad range of seasonal variation in precipitation and temperature (Supplementary Fig. 1), and a wide range of grassland types (Supplementary Data 1). In our analysis, we treated each 1 m² subplot as a "community" and the replicated subplots within a site as the "larger scale" sensu Whittaker[28]. We computed diversity, asynchrony, and stability within a community (local "alpha" scale) and across the three replicated communities within a site (larger "gamma" scale) (see "Methods"). We then used bivariate analysis and structural equation modeling (SEM)[29] to assess fertilizer impacts, and disentangle the relative contributions of diversity and asynchrony to stability (Fig. 1e).

## Results and discussion
**Fertilization effects on diversity, asynchrony, and stability.**
Analyses of variance revealed the negative effects of nutrient inputs on biodiversity and stability at the two scales investigated, consistent with recent findings from a single site[26]. Fertilization consistently reduced species richness, alpha, and gamma stability, but had no effect on beta diversity (Supplementary Fig. 2). Bivariate analyses further revealed the negative effects of nutrient inputs on biodiversity–stability relationships at the two scales investigated (Fig. 2). Relationships were generally consistent across the different periods of experimental duration considered (Supplementary Table 1). Under ambient (unfertilized) conditions, species richness was positively associated with alpha and gamma stability (Fig. 2a, b), but fertilization weakened the positive effect of species richness on stability at the two scales (Fig. 2c, d). Fertilization reduced local stability of grassland functioning by increasing temporal variability in species-rich communities (Supplementary Fig. 3). Similarly, high beta diversity (variation in species composition among communities) was positively associated with spatial asynchrony and gamma stability under ambient conditions (Fig. 2e, f), but again fertilization weakened the positive effect of beta diversity on spatial asynchrony and gamma stability (Fig. 2g, h). These results remained when

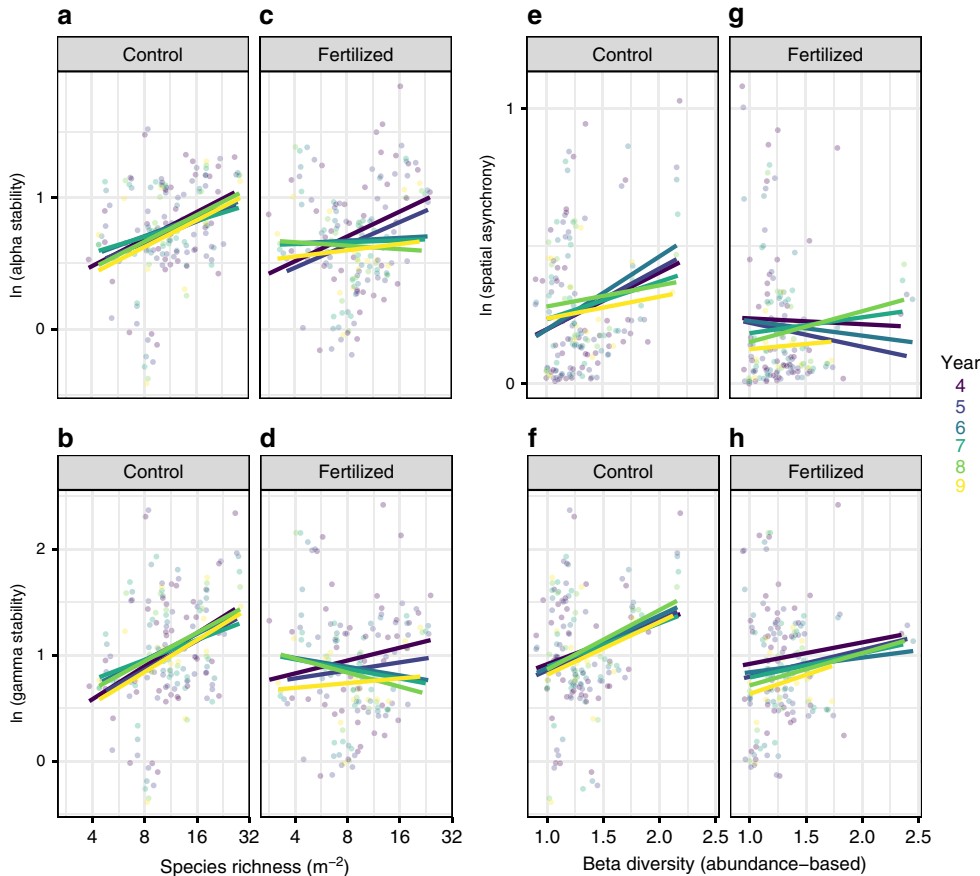

**Fig. 2 Impact of fertilization on biodiversity–stability relationships across spatial scales.** Stability was measured as the temporal mean of primary productivity divided by its temporal standard deviation. Relationships were generally consistent among the periods of experimental duration considered (Supplementary Table 1). Species richness was positively associated with **a** alpha (slope and 95% CIs across time = 0.17 (0.08–0.26)) and **b** gamma stability (0.27 (0.15–0.39)) in the unmanipulated communities, but unrelated to **c** alpha (0.01 (−0.07 to 0.10)) and **d** gamma stability (−0.02 (−0.09 to 0.14)) in the fertilized communities. Beta diversity was positively related to **e** spatial asynchrony (0.18 (0.06–0.30)) and **f** gamma stability (0.47 (0.19–0.74)) in the unmanipulated communities, but unrelated to **g** spatial asynchrony (−0.01 (−0.13 to 0.12)) and **h** gamma stability (0.21 (−0.07 to 0.50)) in the fertilized communities. Note the scale of y-axis differ across panels and this needs to be considered when visually inspecting slopes. Each dot represents the collective subplots across the three replicated 1-m$^2$ subplots for each site, treatment and duration period (n = 160). Colors represent the periods of experimental duration.

accounting for variation in climate using residual regression (Supplementary Fig. 4), when using local diversity indices accounting for species abundance (Supplementary Fig. 5), and when data were divided into overlapping intervals of 4 years (Supplementary Fig. 6). Our results extend previous evidence of the negative impact of fertilization on the diversity–stability relationship obtained within local plots and over shorter experimental periods[4,6,26]. Importantly, they show that these negative effects propagate from within to among communities. To our knowledge, our study is the first to report the negative impacts of fertilization on the relationships of beta diversity, with spatial asynchrony and gamma stability.

**Mechanisms linking diversity and stability.** To understand the relative role of local vs. larger scale community properties in determining asynchrony and stability at different spatial scales, we conducted SEM analyses, including all measures in a single causal model (Fig. 3, Supplementary Fig. 7 and Supplementary Table 2). Under ambient conditions, SEM revealed that higher plant species richness contributed to greater alpha and gamma stability largely through higher asynchronous dynamics among species (species asynchrony, standardized path coefficient = 0.39), and not necessarily through greater species stability

(standardized path coefficient = 0.01; Fig. 3a and Supplementary Fig. 8a, b). The positive association between species richness and alpha stability is consistent with existing experimental[17,24] and shorter-term observational evidence[4,30,31]. Our results confirm that the stabilizing effects of species richness in naturally assembled grassland communities is largely driven by species asynchrony, but not species stability[4,6,22,26]. In addition, they show that the positive impact of species richness on the stability of community productivity via species asynchrony in turn leads to greater stability of productivity at the larger spatial scale.

While correlated with species richness, higher beta diversity also contributed to greater gamma stability through an independent pathway, namely via higher asynchronous dynamics among local communities (spatial asynchrony, standardized path coefficient = 0.20, Fig. 3a). While theoretical studies have suggested a role for beta diversity in driving spatial asynchrony[15,16], previous empirical studies conducted along a nitrogen gradient at a single site[26] or across 62 sites with non-standardized protocols[21] did not find an association between these two variables. Here, we show that the presence of different species among local communities is linked to higher variation in dynamics among them, demonstrating the stabilizing role of beta diversity at larger spatial scales through spatial asynchrony. This also indicates the need for

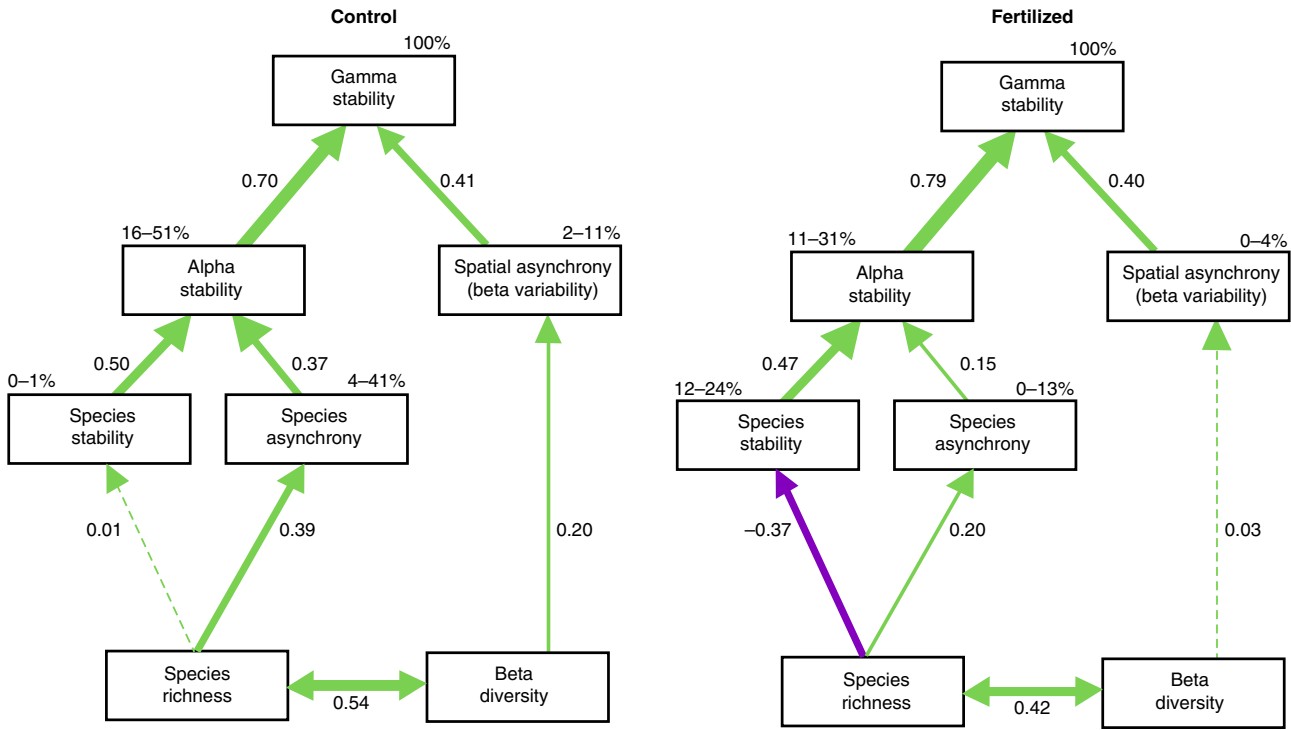

**Fig. 3 Summary of meta-analysis results.** Direct and indirect pathways through which biodiversity, asynchrony, and stability at multiple spatial scales determines gamma stability under **a** unmanipulated control or **b** fertilized condition. Boxes represent measured variables and arrows represent relationships among variables. Numbers next to the arrows are averaged effect sizes as standardized path coefficients. Solid green and purple arrows represent significant ($P \leq 0.05$) positive and negative coefficients, respectively, and dashed green and purple arrows represent nonsignificant coefficients. Widths of paths are scaled by standardized path coefficients. Percentages next to endogenous variables indicate the range of variance explained by the model ($R^2$) across period of experimental duration.

**Table 2 Summary of meta-analysis results showing tests for differences of model paths between the unmanipulated control and fertilized conditions, including Cochrane $Q$ statistics for the treatment effect (unmanipulated control vs. fertilized condition) with associated degrees of freedom and $P$ values.**

| Pathway | Cochrane $Q$ statistics | d.f. | $P$ value |
|---|---|---|---|
| Within communities | | | |
| Species richness → species stability | 36.52 | 1 | <0.001 |
| Species richness → species asynchrony | 3.44 | 1 | 0.064 |
| Species stability → alpha stability | 0.09 | 1 | 0.77 |
| Species asynchrony → alpha stability | 7.15 | 1 | 0.008 |
| Among communities | | | |
| Beta diversity → spatial asynchrony | 4.52 | 1 | 0.034 |
| Alpha stability → gamma stability | 5.27 | 1 | 0.022 |
| Spatial asynchrony → gamma stability | 0.11 | 1 | 0.74 |

multisite replication with standardized treatments and protocols to detect such effects.

Importantly, fertilization acted to destabilize productivity at the local and larger spatial scale through several mechanisms (Fig. 3 and Table 2). At the local scale, fertilization weakened the positive effects of plant species richness on alpha and gamma stability (Fig. 2a–d) via a combination of two processes (Fig. 3b and Supplementary Fig. 8c, d). First, the positive relationship between species richness and species asynchrony in the control communities (standardized path coefficient = 0.39, Fig. 3a), was weaker in the fertilized communities (standardized path coefficient = 0.20, Fig. 3b). Moreover, this general positive effect of richness on asynchrony was counteracted by a second stronger negative relationship of richness with species stability (standardized

path coefficient = −0.37). Such negative effect of fertilization on species stability was not observed under ambient conditions, and could be due to shifts in functional composition in species-rich communities from more stable conservative species to less stable exploitative species in a temporally variable environment[32,33]. Together, these two effects explain the overall weaker alpha stability at higher richness with fertilization. We did not find evidence that the loss of diversity caused by fertilization (an average of −1.8 ± 0.5 species m$^{-2}$, Supplementary Fig. 2a and Supplementary Fig. 9a) was related to the decline of alpha stability, confirming results from other studies[5,6] and earlier NutNet results[4] obtained over shorter time periods. This could be because the negative feedback of the loss of richness caused by fertilization on stability requires a longer experimental duration,

or greater loss of plant diversity, to manifest[9,34]. Another possible explanation is that fertilization may have a direct positive effect on stability, by increasing community biomass ($t = 2.41$, d.f. = 326, $P = 0.016$) and enhancing stability via overyielding effects[35], a formal test that would require monocultures.

At the larger scale, fertilization reduced the strength of the relationship between beta diversity and gamma stability by reducing the strength of the relationship between beta diversity and spatial asynchrony (standardized path coefficient = 0.20 in Fig. 3a vs. standardized path coefficient = 0.03 in Fig. 3b). This result provides evidence that fertilization can reduce the stabilizing role of spatial asynchrony among initially dissimilar communities. We did not find evidence that this was due to a negative feedback of changes in beta diversity caused by fertilization on gamma stability (Supplementary Fig. 2b and Supplementary Fig. 9b). The positive relationship between beta diversity and spatial asynchrony, and the negative impact of fertilization on that relationship, suggests that the spatial insurance effect caused by variation in species composition among local communities may be disrupted in a eutrophic world.

**Implications**. Our results support the idea that asynchronous dynamics among species in species-rich communities play a stabilizing role, and show that this effect propagates to larger spatial scales[21,26]. Furthermore, to our knowledge, our study is the first to report the positive association between beta diversity and gamma stability through spatial asynchrony in real-world grasslands. Importantly, fertilization reduced the contribution of biodiversity to these stabilizing mechanisms at both scales, diminishing the local and spatial insurance of biodiversity on stability. Such diminished insurance effects lead to a reduced ecosystem stability at larger scales. Future climate will be characterized by more variability, including more frequent extreme events[13]. Our results indicate that preserving ecosystem stability across spatial scales in a changing world requires conserving biodiversity within and among local communities. Moreover, policies and management procedures that prevent and mitigate eutrophication are needed to safeguard the positive effects of biodiversity on stability at multiple scales.

## Methods

**Study sites and experimental design**. The study sites are part of the NutNet experiment (Supplementary Data 1; http://nutnet.org/)[27]. Plots at each site are 5 × 5 m separated by at least 1 m. All sites included in the analyses presented here included unmanipulated plots and fertilized plots with nitrogen (N), phosphorus (P), and potassium and micronutrients (K) added in combination (NPK$_+$). N, P, and K were applied annually before the beginning of the growing season at rates of 10 g m$^{-2}$ y$^{-1}$. N was supplied as time-release urea ((NH$_2$)$_2$CO) or ammonium nitrate (NH$_4$NO$_3$). P was supplied as triple super phosphate (Ca(H$_2$PO$_4$)$_2$), and K as potassium sulfate (K$_2$SO$_4$). In addition, a micronutrient mix (Fe, S, Mg, Mn, Cu, Zn, B, and Mo) was applied at 100 g m$^{-2}$ y$^{-1}$ to the K-addition plots, once at the start of the experiment, but not in subsequent years to avoid toxicity. Treatments were randomly assigned to the 25 m$^2$ plots and were replicated in three blocks at most sites (some sites had fewer/more blocks or were fully randomized). Sampling was done in 1 m$^2$ subplots and followed a standardized protocol at all sites[27].

**Site selection**. Data were retrieved on 1 May 2020. To keep a constant number of communities per site and treatment, we used three blocks per site, excluding additional blocks from sites that had more than three (Supplementary Data 1). Sites spanned a broad envelope of seasonal variation in precipitation and temperature (Supplementary Fig. 1), and represent a wide range of grassland types, including alpine, desert and semiarid grasslands, prairies, old fields, pastures, savanna, tundra, and shrub-steppe (Supplementary Data 1).

Stability and asynchrony measurements are sensitive to taxonomic inconsistencies. We adjusted the taxonomy to ensure consistent naming over time within sites. This was usually done by aggregating taxa at the genus level when individuals were not identified to species in all years. Taxa are however referred to as "species".

We selected sites that had a minimum of 4 years, and up to 9 years of posttreatment data. Treatment application started at most sites in 2008, but some sites started later resulting in a lower number of sites with increasing duration of the study, from 42 sites with 4 years of posttreatment duration to 15 sites with 9 years of duration (Supplementary Data 1). Longer time series currently exist, but for a limited number of sites within our selection criteria.

**Primary productivity and cover**. We used aboveground live biomass as a measure of primary productivity, which is an effective estimator of aboveground net primary production in herbaceous vegetation[36]. Primary productivity was estimated annually by clipping at ground level all aboveground live biomass from two 0.1 m$^2$ (10 × 100 cm) quadrats per subplot. For shrubs and subshrubs, leaves and current year's woody growth were collected. Biomass was dried to constant mass at 60 °C and weighed to the nearest 0.01 g. Areal percent cover of each species was measured concurrently with primary productivity in one 1 × 1 m subplot, in which no destructive sampling occurred. Cover was visually estimated annually to the nearest percent independently for each species, so that total summed cover can exceed 100% for multilayer canopies. Cover and primary productivity were estimated twice during the year at some sites with strongly seasonal communities. This allowed to assemble a complete list of species and to follow management procedures typical of those sites. For those sites, the maximum cover of each species and total biomass were used in the analyses.

**Diversity, asynchrony, and stability across spatial scales**. We quantified local scale and larger-scale diversity indices across the three replicated 1-m$^2$ subplots for each site, treatment and duration period using cover data[37,38]. In our analysis, we treated each subplot as a "community" and the collective subplots as the "larger scale" sensu Whittaker[28]. Local scale diversity indices (species richness, species evenness, Shannon, and Simpson) were measured for each community, and averaged across the three communities for each treatment at each site resulting in one single value per treatment and site. Species richness is the average number of plant species. Shannon is the average of Shannon–Weaver indices[39]. Species evenness is the average of the ratio of the Shannon–Weaver index and the natural logarithm of average species richness (i.e., Pielou's evenness[40]). Simpson is the average of inverse Simpson indices[41]. Due to strong correlation between species richness and other common local diversity indices (Shannon: $r = 0.90$ (95% confidence intervals (CIs) = 0.87–0.92), Simpson: $r = 0.88$ (0.86–0.91), Pielou's evenness: $r = 0.62$ (0.55–0.68), with d.f. = 324 for each), we used species richness as a single, general proxy for those variables in our models. Results using these diversity indices did not differ quantitatively from those presented in the main text using species richness (Supplementary Fig. 5), suggesting that fertilization modulate diversity effects largely through species richness. Following theoretical models[15,16], we quantified abundance-based gamma diversity as the inverse Simpson index over the three subplots for each treatment at each site and abundance-based beta diversity, as the multiplicative partitioning of abundance-based gamma diversity: abundance-based beta equals the abundance-based gamma over Simpson[28,42], resulting in one single beta diversity value per treatment and site. We used abundance-based beta diversity index because it is directly linked to ecosystem stability in theoretical models[15,16], and thus directly comparable to theories. We used the R functions "diversity", "specnumber", and "vegdist" from the vegan package[43] to calculate Shannon–Weaver, Simpson, and species richness indices within and across replicated plots.

Stability at multiple scales was determined both without detrending and after detrending data. For each species within communities, we detrended by using species-level linear models of percent cover over years. We used the residuals from each regression as detrended standard deviations to calculate detrended stability[17]. Results using detrended stability did not differ quantitatively from those presented in the main text without detrending. Stability was defined by the temporal invariability of biomass (for alpha and gamma stability) or cover (for species stability and species asynchrony), calculated as the ratio of temporal mean to standard deviation[14,17]. Gamma stability represents the temporal invariability of the total biomass of three plots with the same treatment, alpha stability represents the temporal invariability of community biomass averaged across three plots per treatment and per site, and species stability represents the temporal invariability of species cover averaged across all species and the three plots per treatment[14]. The mathematical formula are:

$$\text{Species stability} = \frac{\sum_{i,k} m_{i,k}}{\sum_{i,k} \sqrt{w_{ii,kk}}}, \tag{1}$$

$$\text{Alpha stability} = \frac{\sum_k \mu_k}{\sum_k \sqrt{v_{kk}}}, \tag{2}$$

$$\text{Gamma stability} = \frac{\sum_k \mu_k}{\sqrt{\sum_{k,l} \nu_{kl}}}, \tag{3}$$

where $m_{i,k}$ and $w_{ii,kk}$ denote the temporal mean and variance of the cover of species $i$ in subplot $k$; $\mu_k$ and $v_{kk}$ denote the temporal mean and variance of community

biomass in subplot $k$, and $v_{kl}$ denotes the covariance in community biomass between subplot $k$ and $l$. We then define species asynchrony as the variance-weighted correlation across species, and spatial asynchrony as the variance-weighted correlation across plots:

$$\text{Species asynchrony} = \frac{\sum_{i,k} \sqrt{w_{ii,kk}}}{\sum_k \sqrt{\sum_{ij,kl} w_{ij,kl}}}, \quad (4)$$

$$\text{Spatial asynchrony} = \frac{\sum_k \sqrt{v_{kk}}}{\sqrt{\sum_{k,l} v_{kl}}}, \quad (5)$$

where $w_{ij,kl}$ denotes the covariance in species cover between species $i$ in subplot $k$ and species $j$ in subplot $l$.

These two asynchrony indices quantify the incoherence in the temporal dynamics of species cover and community biomass, respectively, which serve as scaling factors to link stability metrics across scales[14] (Fig. 1). To improve normality, stability, and asynchrony measures were logarithm transformed before analyses. We used the R function "var.partition" to calculate asynchrony and stability across spatial scales[14].

**Climate data**. Precipitation and temperature seasonality were estimated for each site, using the long-term coefficient of variation of precipitation (MAP_VAR) and temperature (MAT_VAR), respectively, derived from the WorldClim Global Climate database (version 1.4; http://www.worldclim.org/)[44].

**Analyses**. All analyses were conducted in R 4.0.2 (ref. [45]) with $N = 42$ for each analysis unless specified. First, we used analysis of variance to determine the effect of fertilization, and period of experimental duration on biodiversity and stability at the two scales investigated. Models including an autocorrelation structure with a first-order autoregressive model (AR(1)), where observations are expected to be correlated from 1 year to the next, gave substantial improvement in model fit when compared with models lacking autocorrelation structure. Second, we used bivariate analyses and linear models to test the effect of fertilization and period of experimental duration on biodiversity–stability relationships at the two scales investigated. Again, models including an autocorrelation structure gave substantial improvement in model fit (Supplementary Table 1)[46–48]. We ran similar models based on nutrient-induced changes in diversity, stability, and asynchrony. For each site, relative changes in biodiversity, stability, and asynchrony at the two scales considered were calculated, as the natural logarithm of the ratio between the variable in the fertilized and unmanipulated plots (Supplementary Fig. 9). Because plant diversity, asynchronous dynamics, and temporal stability may be jointly controlled by interannual climate variability[22], we ran similar analyses on the residuals of models that included the coefficient of variation among years for each of temperature and precipitation. Results of our analyses controlling for inter-annual climate variability did not differ qualitatively from the results presented in the text (Supplementary Fig. 4). In addition, to test for temporal trends in stability and diversity responses to fertilization, we used data on overlapping intervals of four consecutive years. Results of our analyses using temporal trends did not differ qualitatively from the results presented in the text (Supplementary Fig. 6). Inference was based on 95% CIs.

Second, we used SEM[29] with linear models, to evaluate multiple hypothesis related to key predictions from theories (Table 1). The path model shown in Fig. 1e was evaluated for each treatment (control and fertilized), and we ran separate SEMs for each period of experimental duration (from 4 to 9 years of duration). We generated a summary SEM by performing a meta-analysis of the standardized coefficients across all durations for each treatment. We then tested whether the path coefficients for each model differed by treatment by testing for a model-wide interaction with the "treatment" factor. A positive interaction for a given path implied that effects of one variable on the other are significantly different between fertilized and unfertilized treatments. We used the R functions "psem" to fit separate piecewise SEMs[49] for each duration and combined the path coefficients from those models, using the "metagen" function[50].

**Reporting summary**. Further information on research design is available in the Nature Research Reporting Summary linked to this article.

## Data availability

The data that support the findings of this study are available via GitHub (https://github.com/YannHautier/NutNetStabilityScaleUp). Data sources are provided with this paper. WorldClim global climate database is freely available through the World Data Center for Climate (WDCC; cera-www.dkrz.de), as well as through the CCAFS-Climate data portal (http://ccafs-climate.org).

## Code availability

R code of all analyses are available via https://github.com/YannHautier/NutNetStabilityScaleUp.

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

## Acknowledgements

The research leading to these results has received funding from the European Union. Seventh Framework Programme (FP7/2007-2013) under grant agreement no. 298935 to Y.H. (with A.H. and E.W.S.). This work was generated using data from the Nutrient Network collaborative experiment, funded at the site scale by individual researchers and coordinated through Research Coordination Network funding from NSF to E.B. and E.W.S. (grant #DEB-0741952). Nitrogen fertilizer was donated to the Nutrient Network by Crop Production Services, Loveland, CO. We acknowledge support from the LTER Network Communications Office and DEB-1545288. M.L. was supported by the TULIP Laboratory of Excellence (ANR-10-LABX-41), and by the BIOSTASES Advanced Grant funded by the European Research Council under the European Union's Horizon 2020 research and innovation programme (grant agreement no. 666971). S.W. was supported by the National Natural Science Foundation of China (31988102). We thank Rita S. L. Veiga and George A. Kowalchuk for suggestions that improved the manuscript.

## Author contributions

Y.H., P.Z., K.R.W., M.L., and S.W. developed and framed research questions. Y.H. and S.W. analyzed the data with help from P.Z., K.R.W., E.W.S., J.E.K.B., S.E.K., K.J.K., and J.S.L. Y.H. wrote the paper with contributions and input from all authors. E.W.S. and E.T.B. are Nutrient Network coordinators. The author contribution matrix is provided as Supplementary Data 2.

## Competing interests

The authors declare no competing interests.

## Additional information

Yann Hautier [1✉], Pengfei Zhang[1,2,3,4], Michel Loreau[5], Kevin R. Wilcox[6], Eric W. Seabloom [7], Elizabeth T. Borer [7], Jarrett E. K. Byrnes[8], Sally E. Koerner[9], Kimberly J. Komatsu [10], Jonathan S. Lefcheck [11], Andy Hector [12], Peter B. Adler[13], Juan Alberti[14], Carlos A. Arnillas[15], Jonathan D. Bakker [16], Lars A. Brudvig[17], Miguel N. Bugalho[18], Marc Cadotte[19], Maria C. Caldeira [20], Oliver Carroll[21], Mick Crawley[22], Scott L. Collins[23], Pedro Daleo [14], Laura E. Dee[24], Nico Eisenhauer [25,26], Anu Eskelinen [25,27,28], Philip A. Fay[29], Benjamin Gilbert[30], Amandine Hansar[31], Forest Isbell [7], Johannes M. H. Knops[32], Andrew S. MacDougall[21], Rebecca L. McCulley [33], Joslin L. Moore [34], John W. Morgan[35], Akira S. Mori [36], Pablo L. Peri[37], Edwin T. Pos[1], Sally A. Power [38], Jodi N. Price [39], Peter B. Reich [38,40], Anita C. Risch [41], Christiane Roscher [25,42], Mahesh Sankaran [43,44], Martin Schütz[40], Melinda Smith [45,46],

Carly Stevens [47], Pedro M. Tognetti [48], Risto Virtanen [28], Glenda M. Wardle [49], Peter A. Wilfahrt[6] & Shaopeng Wang [50 ✉]

[1]Ecology and Biodiversity Group, Department of Biology, Utrecht University, Padualaan 8, 3584 CH Utrecht, The Netherlands. [2]State Key Laboratory of Grassland and Agro-Ecosystems, School of Life Science, Lanzhou University, 730000 Lanzhou, Gansu Province, People's Republic of China. [3]Institute of Eco-Environmental Forensics of Shandong University, 266237 Jinan, Shandong Province, People's Republic of China. [4]Ministry of Justice Hub for Research & Practice in Eco-Environmental Forensics, 266237 Qingdao, Shandong Province, People's Republic of China. [5]Centre for Biodiversity Theory and Modelling, Theoretical and Experimental Ecology Station, CNRS, 2 route du CNRS, 09200 Moulis, France. [6]Department of Ecosystem Science and Management, University of Wyoming, Laramie, WY, USA. [7]Department of Ecology, Evolution, and Behavior, University of MN, St. Paul, MN 55108, USA. [8]Department of Biology, University of Massachusetts Boston, Boston, MA 02125, USA. [9]Department of Biology, University of North Carolina Greensboro, Greensboro, NC, USA. [10]Smithsonian Environmental Research Center, Edgewater, MD 21037, USA. [11]Tennenbaum Marine Observatories Network, MarineGEO, Smithsonian Environmental Research Center, Edgewater, MD 21037, USA. [12]University of Oxford Department of Plant Sciences, Oxford OX1 3RB, UK. [13]Department of Wildland Resources and the Ecology Center, Utah State University, Logan, UT 84322, USA. [14]Instituto de Investigaciones Marinas y Costeras (IIMyC), FCEyN, UNMdP-CONICET, CC 1260 Correo Central, B7600WAG Mar del Plata, Argentina. [15]Department of Physical and Environmental Sciences, University of Toronto at Scarborough, Scarborough, ON, Canada. [16]School of Environmental and Forest Sciences, University of Washington, Seattle, WA 98195-4115, USA. [17]Department of Plant Biology and Program in Ecology, Evolutionary Biology, and Behavior, Michigan State University, East Lansing, MI, USA. [18]Centre for Applied Ecology "Prof. Baeta Neves" (CEABN-InBIO), School of Agriculture, University of Lisbon, Lisbon, Portugal. [19]Department of Biological Sciences, University of Toronto at Scarborough, Scarborough, ON, Canada. [20]Forest Research Centre, School of Agriculture, University of Lisbon, Lisbon, Portugal. [21]Department of Integrative Biology, University of Guelph, Guelph, ON N1G2W1, Canada. [22]Life Sciences, Imperial College London, Silwood Park, Ascot SL5 7PY, UK. [23]University of New Mexico, Department of Biology, Albuquerque, NM 87131, USA. [24]Department of Ecology and Evolutionary Biology, University of Colorado at Boulder, 1560 30th Street, Boulder, CO 80309-0450, USA. [25]German Centre for Integrative Biodiversity Research (iDiv) Halle-Jena-Leipzig, Deutscher Platz 5e, 04103 Leipzig, Germany. [26]Leipzig University, Institute of Biology, Deutscher Platz 5e, 04103 Leipzig, Germany. [27]Department of Physiological Diversity, Helmholtz Centre for Environmental Research - UFZ, Leipzig, Germany. [28]Department of Ecology and Genetics, University of Oulu, Oulu, Finland. [29]USDA-ARS Grassland, Soil, and Water Research Laboratory, Temple, TX 76502, USA. [30]Department of Ecology and Evolutionary Biology, University of Toronto, Toronto, ON M5S3B2, Canada. [31]Centre de recherche en écologie expérimentale et prédictive (CEREEP-Ecotron IleDeFrance), Département de biologie, Ecole normale supérieure, CNRS, PSL University, 77140 St-Pierre-les-Nemours, France. [32]Department of Heatth and Environmental Sciences, Xi'an Jiaotong liverpool University, 214123 Suzhou, Jiangsu, China. [33]University of Kentucky, Plant & Soil Science, 1405 Veterans Drive, Lexington, KY 40546-0312, USA. [34]School of Biological Sciences, Monash University, Clayton Campus, Clayton, VIC 3800, Australia. [35]Department of Ecology, Environment & Evolution, La Trobe University, Bundoora, VIC 3086, Australia. [36]Graduate School of Environment and Information Sciences, Yokohama National University, 79-7 Tokiwadai, Hodogaya, Yokohama, Kanagawa 240-8501, Japan. [37]INTA (National Institute of Agricultural Research)- UNPA (Southern Patagonia National University)-CONICET, Santa Cruz, Argentina. [38]Hawkesbury Institute for the Environment, Western Sydney University, Locked Bag 1797, Penrith, NSW 2751, Australia. [39]Institute of Land, Water and Society, Charles Sturt University, Albury, NSW 2640, Australia. [40]Department of Forest Resources, University of Minnesota, Saint Paul, MN, USA. [41]Swiss Federal Institute for Forest, Snow and Landscape Research WSL, Zuercherstrasse 111, 8903 Birmensdorf, Switzerland. [42]UFZ, Helmholtz Centre for Environmental Research, Physiological Diversity, Permoserstrasse 15, 04318 Leipzig, Germany. [43]Ecology & Evolution Group, National Centre for Biological Sciences, TIFR, Bangalore, Karnataka 560065, India. [44]School of Biology, University of Leeds, Leeds LS2 9JT, UK. [45]Department of Biology, Colorado State University, Fort Collins, CO 80523, USA. [46]Graduate Degree Program in Ecology, Colorado State University, Fort Collins, CO 80523, USA. [47]Lancaster Environment Centre, Lancaster University, Lancaster LA1 4YQ, UK. [48]IFEVA-Facultad de Agronomia, Universidad de Buenos Aires - CONICET, Av San Martin 4453, C1417DSE Ciudad Autonoma de Buenos Aires, Argentina. [49]School of Life and Environmental Sciences, University of Sydney, Sydney, NSW 2006, Australia. [50]Institute of Ecology, College of Urban and Environmental Science, and Key Laboratory for Earth Surface Processes of the Ministry of Education, Peking University, 100871 Beijing, China. ✉email: y.hautier@uu.nl; shaopeng.wang@pku.edu.cn

