## [Peer Review File · Nature Communications]

Reviewers' Comments:

Reviewer #1:

Remarks to the Author:

I have read the manuscript NCOMMS-20-25181 by Hautier et al. with great interest. The paper is really well written with clear paragraphs and language. It is of adequate length and cites the relevant literature. The three main figures provide the most important information and the supplementary figures gives the necessary backup and additional context. I have only some minor comments. The paper gives novel insight into how biodiversity begets stability at multiple scales, but only under non-eutrophicated conditions. The statistical analyses are valid and properly carried out, and the data will be made publicly available. I strongly encourage the authors to also make the computer code available.

It is not often one gets the privilege of reviewing a manuscript that is almost ready for publishing without a few major edits. Well done.

Minor comments:

- Fig. 2 and similar supplementary figures: perhaps consider using a continuous color scale that makes it more intuitive which color matches which year. Like the Viridis scale.
- Line 182: communities
- Line 303-306: Can the authors provide information about how frequently individuals could not be determined to the species level? Were individuals frequently referred to relatively species-rich genera? I do not think that sometimes aggregating to the genus level will have a big impact on the results, but some reflection on this matter would be informative. Perhaps some other papers using the NutNet data have done sensitivity analyses?
- Methods: please use either "subplot" or "patch" consistently, not both. Unless there is a clear rationale for using both.

Reviewer #2:

Remarks to the Author:

This paper applies a recent partitioning framework to identify how species dynamics within- and across-communities scale to effect ecosystem stability under unfertilized and fertilized conditions. It uses data from the NutNet experiment, which applied the same replicated treatments across 41 sites with at least 4 years of data (up to 9 years). It found that a positive relationship between species diversity, species asynchrony, and local stability in unfertilized plots, as well as a positive relationship between beta diversity, spatial asynchrony, and gamma stability in unfertilized plots. In control, fertilization weakened these diversity stabilizing mechanisms. Overall this paper is an elegant application of a new framework to a robust dataset, well written with results of broad interest.

I would like to see some comparison of the overall stability of unfertilized and fertilized plots. While the paper focused on mechanisms, it seemed from Fig 2 that average stability might be about the same, with fertilization potentially increasing stability at low diversity/reducing stability at high diversity.

148 – minor, but "habitat heterogeneity" sounded like a spatial process, would rephrase

175 – given the emphasis on temporal dynamics, would like to see a histogram of time series length in the supplement (if most sites are near the four year length, would seem a bit short for the type of analyses)

203 – nice

234 – need to demonstrate that productivity was actually less stable under fertilization. If it wasn't, the wording here feels a bit disingenuous, and it should be reframed that the mechanisms of stability changed with fertilization.

240 – this was hard to follow, changing “however” to “moreover” might help. Or perhaps rephrasing as “in addition to weakening this positive effect of richness on asynchrony, fertilization strengthened a negative relationship between...”

254 – can this be tested within the framework? if not, some discussion of limitations would be helpful

Fig 1 – this didn't represent well in grayscale, increase shade contrast for the communities (I also found the CA, CB, etc on panel d cluttered – think it is clear enough from the among communities graphic)

Fig 3 – also didn't represent well in grayscale

Reviewer #3:

Remarks to the Author:

This is a very interesting paper that uses a global long-term experimental dataset examine how fertilization impacts the stability of biomass production in grassland plant communities. This is an important paper showing how fertilization weakens the role of community diversity as a mechanism stabilizing productivity.

As requested by the editor, my review comments are primarily focused on the SEM models. I have, however, read the whole paper and generally endorse the methods used in this paper, and the conclusions reached.

L412. The SEM methodology used is reasonable. Given, however, that there are only two treatments, I wonder whether a multigroup model could be an effective approach. Much of the impact of this paper rests on the conclusion that path coefficients differed between the fertilized and unfertilized treatments. It would be nice to have a formal test of whether these relationships differ.

Nowhere in the main text or supplements can I find information on SEM model fit (e.g. chi square tests, CFI or other indices). It is important that an indication of adequate model fit be provided. Given the number of models tested, I would expect that not all would have strong fit, but at least the majority need to have good fit. Ideally, an extended data table detailing these for each model would be provided.

Minor points:

L289. Not sure why it is specified that micronutrients were added to K plots, as the rest of the description indicates that it is control and NPK plots being used in the study.

L298. How were excluded blocks selected? Data table 1 indicates that this was a judgement based process; some more detail would be useful. This could be added to the caption for Data table 1.

Response to reviews of manuscript ID NCOMMS-20-25181 (Hautier et al.)

> We thank the three reviewers for their helpful and constructive comments that led to new analyses and other changes that we feel have strengthened our paper. Below, we give a detailed point-by-point response (in Calibri prefaced '>') to the three reviews (in Times New Roman).

Review 1

I have read the manuscript NCOMMS-20-25181 by Hautier et al. with great interest. The paper is really well written with clear paragraphs and language. It is of adequate length and cites the relevant literature. The three main figures provide the most important information and the supplementary figures gives the necessary backup and additional context. I have only some minor comments. The paper gives novel insight into how biodiversity begets stability at multiple scales, but only under non-eutrophicated conditions. The statistical analyses are valid and properly carried out, and the data will be made publicly available. I strongly encourage the authors to also make the computer code available.

It is not often one gets the privilege of reviewing a manuscript that is almost ready for publishing without a few major edits. Well done.

> We appreciate that the Reviewer is very enthusiastic about our work and believes the paper suits the broad readership of Nature Communications. We will make the R scripts available as well as the data.

Minor comments:

- Fig. 2 and similar supplementary figures: perhaps consider using a continuous color scale that makes it more intuitive which color matches which year. Like the Viridis scale.

> We thank the reviewer for helping us making our figures more readable for a majority of readers. We now used the Viridis scale as suggested for all the figures.

- Line 182: communities

> Changed as suggested

- Line 303-306: Can the authors provide information about how frequently individuals could not be determined to the species level? Were individuals frequently referred to relatively species-rich genera? I do not think that sometimes aggregating to the genus level will have a big

impact on the results, but some reflection on this matter would be informative. Perhaps some other papers using the NutNet data have done sensitivity analyses?

> We added in Extended Table 1 the number of taxa that were classified at the genus level as well as the percentage relative to the total number of species per site (gamma diversity). Number: range = 0-6, mean = 0.9. Percentage: range = 0-36.1%, mean = 5.9%. We conducted a sensitivity analysis by running all analyses with and without aggregating. Results did not differ quantitatively.

- Methods: please use either “subplot” or “patch” consistently, not both. Unless there is a clear rationale for using both.

> Thanks for pointing out at this inconsistency. We modified as suggested by using “subplot” consistently.

Reviewer #2 (Remarks to the Author):

This paper applies a recent partitioning framework to identify how species dynamics within- and across-communities scale to effect ecosystem stability under unfertilized and fertilized conditions. It uses data from the NutNet experiment, which applied the same replicated treatments across 41 sites with at least 4 years of data (up to 9 years). It found that a positive relationship between species diversity, species asynchrony, and local stability in unfertilized plots, as well as a positive relationship between beta diversity, spatial asynchrony, and gamma stability in unfertilized plots. In control, fertilization weakened these diversity stabilizing mechanisms. Overall this paper is an elegant application of a new framework to a robust dataset, well written with results of broad interest.

> We appreciate that the reviewer found our approach elegant and robust with results of broad interest.

I would like to see some comparison of the overall stability of unfertilized and fertilized plots. While the paper focused on mechanisms, it seemed from Fig 2 that average stability might be about the same, with fertilization potentially increasing stability at low diversity/reducing stability at high diversity.

> We added this comparison in Figure S2 and in the results section. These analyses revealed that while fertilization increased stability at low diversity and reduced stability at high diversity, on average the effect of fertilization was to reduce both alpha and gamma stability.

148 – minor, but “habitat heterogeneity” sounded like a spatial process, would rephrase

> We rephrased to ‘niche dimensionality’.

175 – given the emphasis on temporal dynamics, would like to see a histogram of time series length in the supplement (if most sites are near the four year length, would seem a bit short for the type of analyses)

> We thank the reviewer for this suggestion that we approached in two ways. First, we included the most recent data available (May 2020 instead of November 2019). This led to a total number of sites of 42 instead of 41 and increased the number of sites with longer periods of experimental duration. Results are consistent with the ones previously reported, confirming the robustness of our results. We updated all the figures and statistics. Second, to limit the number of supplementary figures, we added the period of experimental duration in Table S1. We now have 15 sites with 9 years, 18 with 8 years, 24 with 7 years, 26 with 6 years, 38 with 5 years, and 42 with 4 years. More than half of our sites (24 out of 42) have at least 7 year data, which is of a reasonable length for stability analysis. However, we can add such a histogram if the reviewer finds it more relevant.

203 – nice

> Thanks

234 – need to demonstrate that productivity was actually less stable under fertilization. If it wasn’t, the wording here feels a bit disingenuous, and it should be reframed that the mechanisms of stability changed with fertilization.

> Good point. As discussed above, we added this comparison in Figure S2 and in the results section. Productivity was indeed less stable with fertilization at the two scales investigated.

240 – this was hard to follow, changing “however” to “moreover” might help. Or perhaps rephrasing as “in addition to weakening this positive effect of richness on asynchrony, fertilization strengthened a negative relationship between...”

> We felt changing ‘however’ to ‘moreover’ would work best in this case.

254 – can this be tested within the framework? if not, some discussion of limitations would be helpful

> Indirect test shows that fertilization increased the temporal mean of productivity, suggesting

an indirect positive effect of fertilization on stability via overyielding. However, a formal test would require monocultures. We added the indirect test and specified, ‘, a test that would require monocultures’.

Fig 1 – this didn’t represent well in grayscale, increase shade contrast for the communities (I also found the CA, CB, etc on panel d cluttered – think it is clear enough from the among communities graphic)

Fig 3 – also didn’t represent well in grayscale

> We are now using the Viridis scale as suggested by Reviewer 1 for each figure including Figure 1 to increase shade contrast. We also removed the cluttered legends inside the panels as suggested.

Reviewer #3 (Remarks to the Author):

This is a very interesting paper that uses a global long-term experimental dataset examine how fertilization impacts the stability of biomass production in grassland plant communities. This is an important paper showing how fertilization weakens the role of community diversity as a mechanism stabilizing productivity.

As requested by the editor, my review comments are primarily focused on the SEM models. I have, however, read the whole paper and generally endorse the methods used in this paper, and the conclusions reached.

> We appreciate that the reviewer found the methods and conclusions suitable.

L412. The SEM methodology used is reasonable. Given, however, that there are only two treatments, I wonder whether a multigroup model could be an effective approach. Much of the impact of this paper rests on the conclusion that path coefficients differed between the fertilized and unfertilized treatments. It would be nice to have a formal test of whether these relationships differ.

> This is an excellent suggestion, however, piecewise structural equation modelling does not yet support a formal multigroup analysis (although at least two approaches are under development). However, testing whether each path coefficient varies by group is philosophically analogous to an interaction between each term in the model and the grouping factor: if the path coefficient differs among groups, then it follows that the interaction should be signifi-

cant. Therefore, we manually tested for a model-wide interaction by treatment (fertilized vs unfertilized). This analysis supports that the paths differ between the unmanipulated control and fertilized condition. We added the analysis and modified the method and result section accordingly. We report the results in a new Table 2.

Nowhere in the main text or supplements can I find information on SEM model fit (e.g. chi square tests, CFI or other indices). It is important that an indication of adequate model fit be provided. Given the number of models tested, I would expect that not all would have strong fit, but at least the majority need to have good fit. Ideally, an extended data table detailing these for each model would be provided.

> We added the model fit in the extended figure S6. Indeed, not all model have an adequate fit, but the majority have, especially after adjusting α for multiple testing.

Minor points:

L289. Not sure why it is specified that micronutrients were added to K plots, as the rest of the description indicates that it is control and NPK plots being used in the study.

> We clarified by changing NPK to NPK₊

L298. How were excluded blocks selected? Data table 1 indicates that this was a judgement based process; some more detail would be useful. This could be added to the caption for Data table 1.

> We added a clarification in the caption of extended data table 1. 'That is, where more than 3 blocks were established, we focused on the first three blocks unless the site lead recommended a different set of blocks.'